# k-Support and Ordered Weighted Sparsity for Overlapping Groups: Hardness and Algorithms

**Cong Han Lim**
University of Wisconsin-Madison
clim9@wisc.edu

**Stephen J. Wright**
University of Wisconsin-Madison
swright@cs.wisc.edu

## Abstract

The $k$-support and OWL norms generalize the $\ell_1$ norm, providing better prediction accuracy and better handling of correlated variables. We study the norms obtained from extending the $k$-support norm and OWL norms to the setting in which there are overlapping groups. The resulting norms are in general NP-hard to compute, but they are tractable for certain collections of groups. To demonstrate this fact, we develop a dynamic program for the problem of projecting onto the set of vectors supported by a fixed number of groups. Our dynamic program utilizes tree decompositions and its complexity scales with the treewidth. This program can be converted to an extended formulation which, for the associated group structure, models the $k$-group support norms and an overlapping group variant of the ordered weighted $\ell_1$ norm. Numerical results demonstrate the efficacy of the new penalties.

## 1 Introduction

The use of the $\ell_1$-norm to induce sparse solutions is ubiquitous in machine learning, statistics, and signal processing. When the variables can be grouped into sets corresponding to different explanatory factors, group variants of the $\ell_1$ penalty can be used to recover solutions supported on a small number of groups. When the collection of groups $\mathcal{G}$ forms a partition of the variables (that is, the groups do *not* overlap), the *group lasso penalty* [19]

$$\Omega_{\mathrm{GL}}(x) := \sum_{G \in \mathcal{G}} \|x_G\|_p \tag{1}$$

is often used. In many cases, however, some variables may contribute to more than one explanatory factor, which leads naturally to overlapping-group formulations. Such is the case in applications such as finding relevant sets of genes in a biological process [10] or recovering coefficients in wavelet trees [17]. In such contexts, the standard group lasso may introduce artifacts, since variables that are contained in different numbers of groups are penalized differently. Another approach is to employ the *latent group lasso* [10]:

$$\Omega_{\mathrm{LGL}}(x) := \min_{x,v} \sum_{G \in \mathcal{G}} \|v_G\|_p \quad \text{such that} \quad \sum_{G \in \mathcal{G}} v_G = x, \tag{2}$$

where each $v_G$ is a separate vector of latent variables supported only on the group $G$. The latent group lasso (2) can be written in terms of atomic norms, where the atomic set is

$$\{x : \|x\|_p \le 1, \operatorname{supp}(x) \subseteq G \text{ for some } G \in \mathcal{G}\}.$$

This set allows vectors supported on any one group. The unit ball is the convex hull of this atomic set.

A different way of extending the $\ell_1$-norm involves explicit use of a sparsity parameter $k$. Argyriou et al. [1] introduce the $k$-support norm $\Omega_k$ from the atomic norm perspective. The atoms are the set of $k$-sparse vectors with unit norm, and the unit ball of the norm is thus

$$\operatorname{conv}\left(\{x : \|x\|_p \le 1, |\operatorname{supp}(x)| \le k\}\right). \tag{3}$$

The $k$-support norm with $p = 2$ offers a tighter alternative to the elastic net, and like the elastic net, it has better estimation performance than the $\ell_1$ norm especially in the presence of correlated variables. Another extension of the $\ell_1$ norm is to the OSCAR/OWL/SLOPE norms [5, 20, 4], which order the elements of $x$ according to magnitude before weighing them:

$$\Omega_{\text{OWL}}(x) \coloneqq \sum\nolimits_{i \in [n]} w_i |x_i^{\downarrow}|. \tag{4}$$

where the weights $w_i$, $i = 1, 2, \ldots, n$ are nonnegative and decreasing and $x^{\downarrow}$ denotes the vector $x$ sorted by decreasing absolute value. This family of norms controls the false discovery rate and clusters correlated variables. These norms correspond to applying the $\ell_{\infty}$ norm to a combinatorial penalty function in the framework of Obozinski and Bach [11, 12], and can be generalized by considering different $\ell_p$-norms. For $p = 2$, we have the SOWL norm [18], whose variational form is

$$\Omega_{\text{SOWL}}(x) \coloneqq \tfrac{1}{2} \min_{\eta \in \mathbb{R}_+^n} \sum\nolimits_{i \in [n]} \left( x_i^2 / \eta_i + w_i |\eta_i^{\downarrow}| \right).$$

We will refer to the generalized version of these norms as pOWL norms. The pOWL norms can be viewed as extensions of the $k$-support norms from the atomic norm angle, which we will detail later.

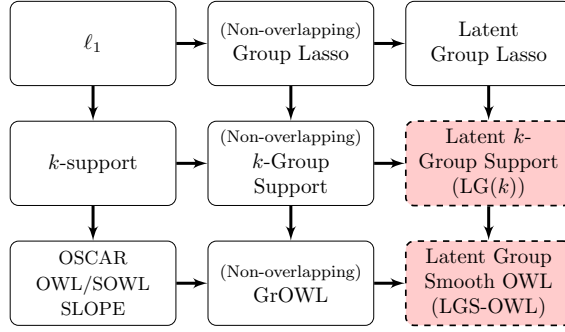

Figure 1: Some sparsity-inducing norms. Each arrow represents an extension of a previous norm. We study the two shaded norms on the right.

In this paper, we study the norms obtained by combining the overlapping group formulations with the $k$-sparse/OWL formulations, with the aim of obtaining the benefits of both worlds. When the groups do not overlap, the combination is fairly straightforward; see the GrOWL norm introduced by Oswal et al. [13]. We consider two classes of norms for overlapping groups. The *latent k-group support (LG(k))* norm, very recently introduced by Rao et al. [15], is defined by the unit ball

$$\text{conv}\left( \left\{ x : \|x\|_p \le 1, \text{supp}(x) \subseteq \bigcup\nolimits_{G \in \mathcal{G}_k} G \text{ for some subset } \mathcal{G}_k \subseteq \mathcal{G} \text{ with } k \text{ groups} \right\} \right), \tag{5}$$

directly extending the $k$-support norm definition to *unions* of groups. We introduce the *latent group smooth OWL (LGS-OWL)* norm, which similarly extends OWL/SOWL/GrOWL. These norms can be applied in the same settings where the latent group lasso has proven to be useful, while adapting better to correlations. We explain how the norms are derived from a combinatorial penalty perspective using the work of Obozinski and Bach [11, 12], and also provide explicit atomic-norm formulations. The LGS-OWL norm can be seen as a combination of $k$-support norms across different $k$.

The rest of this focuses on computational aspects of these norms. Both the LG($k$) norm and the LGS-OWL norm are in general NP-hard to compute. Despite this hardness result, we devise a computational approach that utilizes *tree decompositions* of the underlying *group intersection graph*. The key parameter affecting the efficiency of our algorithms is the *treewidth tw* of the group intersection graph, which is small for certain graph structures such as chains, trees, and cycles. Certain problems with hierarchical groups like image recovery can have a tree structure [17, 3].

Our first main technical contribution is a dynamic program for the *best k-group sparse approximation* problem, which has time complexity $O(2^{O(tw)} \cdot mk + n)$, where $m$ is the total number of groups. For group intersection graphs with a tree structure ($tw = 2$), this leads to a $O(mk + n)$ algorithm, significantly improving on the $O(m^2 k + n)$ algorithm presented in [3]. Next, we build on the principles behind the dynamic program to construct extended formulations of $O(2^{O(tw)} \cdot mk^2 + n)$

size for LG($k$) and $O(2^{O(tw)} \cdot m^3 + n)$ for LGS-OWL, improving by a factor of $k$ or $m$ respectively in the special case in which the tree decomposition is a chain. This approach also yields extended formulations of size $O(nk)$ and $O(n^2)$ for the $k$-support and pOWL norms, respectively. (Previously, only a $O(n^2)$ linear program was known for OWL [5].) We thus facilitate incorporation of these norms into standard convex programming solvers.

**Related Work.** Obozinski and Bach [11, 12] develop a framework for penalties derived by convexifying the sum of a combinatorial function $F$ and an $\ell_p$ term. They describe algorithms for computing the proximal operators and norms for the case of submodular $F$. We use their framework, but note that the algorithms they provide cannot be applied since our functions are not submodular.

Two other works focus directly on sparsity of unions of overlapping groups. Rao et al. [15] introduce the $LG(k)$ norm and approximates it via variable splitting. Baldassarre et al. [3] study the best $k$-group sparse approximation problem, which they prove is NP-hard. For tree-structured intersection graphs, they derive the aforementioned dynamic program with complexity $O(m^2k + n)$.

For the case of $p = \infty$, a linear programming relaxation for the unit ball of the latent $k$-group support norm is provided by Halabi and Cevher [9, Section 5.4]. This linear program is tight if the group-element incidence matrix augmented with an all-ones row is *totally unimodular*. This condition can be violated by simple tree-structured intersection graphs with just four groups.

**Notation and Preliminaries.** Given $A \subseteq [n]$, the vector $x_A$ is the subvector of $x \in \mathbb{R}^n$ corresponding to the index set $A$. For collections of groups $\mathcal{G}$, we use $m$ to denote the number of groups in $\mathcal{G}$, that is, $m = |\mathcal{G}|$. We assume that $\bigcup_{G \in \mathcal{G}} G = [n]$, so that every index $i \in [n]$ appears in at least one group $G \in \mathcal{G}$. The discrete function $C_{\mathcal{G}}(A)$ denotes the minimum number of groups from $\mathcal{G}$ needed to cover $A$ (the smallest set cover).

## 2 Overlapping Group Norms with Group Sparsity-Related Parameters

We now describe the LG($k$) and LGS-OWL norms from the combinatorial penalty perspective by Obozinski and Bach [11, 12], providing an alternative theoretical motivation for the LG($k$) norm and formally motivating and defining LGS-OWL. Given a combinatorial function $F : \{A \subseteq [n]\} \to \mathbb{R} \cup \{+\infty\}$ and an $\ell_p$ norm, a norm can be derived by taking the tightest positively homogeneous convex lower bound of the combined penalty function $F(\mathrm{supp}(x)) + \nu\|x\|_p^p$. Defining $q$ to satisfy $1/p + 1/q = 1$ (so that $\ell_p$ and $\ell_q$ are dual), this procedure results in the norm $\Omega_p^F$, which is given by the convex envelope of the function $\Theta_p^F(x) \coloneqq q^{1/q}(p\nu)^{1/p}F(\mathrm{supp}(x))^{1/q}\|x\|_p$, whose unit ball is

$$\mathrm{conv}\left(\left\{x \in \mathbb{R}^n : \|x\|_p \leq F(\mathrm{supp}(x))^{-1/q}\right\}\right). \tag{6}$$

The norms discussed in this paper can be cast in this framework. Recall that the definition of OWL (4) includes nonnegative weights $w_1 \geq w_2 \geq \ldots w_n \geq 0$. Defining $h : [n] \to \mathbb{R}$ to be the monotonically increasing concave function $h(k) = \sum_{i=1}^k w_i$, we obtain

$$k\text{-support} : F(A) = \begin{cases} 0, & A = \emptyset, \\ 1, & |A| \leq k, \\ \infty, & \text{otherwise}, \end{cases} \qquad \text{LG}(k) \quad : F(A) = \begin{cases} 0, & A = \emptyset, \\ 1, & C_{\mathcal{G}}(A) \leq k, \\ \infty, & \text{otherwise}, \end{cases}$$

$$\text{pOWL} \quad : F(A) = h(|A|), \qquad\qquad \text{LGS-OWL} : F(A) = h(C_{\mathcal{G}}(A)).$$

The definitions of the $k$-support and LG($k$) balls from (3) and (5), respectively, match (6). As for the OWL norms, we can express their unit ball by

$$\mathrm{conv}\left(\bigcup_{i=1}^m \left\{x \in \mathbb{R}^n : \|x\|_p \leq h(i)^{-1/q}, C_{\mathcal{G}}(\mathrm{supp}(x)) = i\right\}\right). \tag{7}$$

This can be seen as taking all of the $k$-support or LG($k$) atoms for each value of $k$, scaling them according to the value of $k$, then taking the convex hull of the resulting set. Hence, the OWL norms can be viewed as a way of interpolating the $k$-support norms across all values of $k$. We take advantage of this interpretation in constructing extended formulations.

**Hardness Results.** Optimizing with the cardinality or non-overlapping group based penalties is straightforward, since the well-known PAV algorithm [2] allows us to exactly compute the proximal operator in $O(n \log n)$ time [12]. However, the picture is different when we allow overlapping groups. There are no fast exact algorithms for overlapping group lasso, and iterative algorithms are typically used. Introducing the group sparsity parameters makes the problem even harder.

**Theorem 2.1.** *The following problems are NP-hard for both* $\Omega_{\mathrm{LG}(k)}$ *and* $\Omega_{\mathrm{LGS\text{-}OWL}}$ *when* $p > 1$:

$$\text{Compute } \Omega(y), \qquad \text{(norm computation)}$$

$$\arg\min_{x \in \mathbb{R}^n} \tfrac{1}{2}\|x - y\|_2^2 \quad \text{such that } \Omega(x) \leq \mu, \qquad \text{(projection operator)}$$

$$\arg\min_{x \in \mathbb{R}^n} \tfrac{1}{2}\|x - y\|_2^2 + \lambda\Omega(x). \qquad \text{(proximal operator)}$$

Therefore, other problems that incorporate these norm are also hard. Note that even if we only allow each element to be in at most two groups, the problem is already hard. We will show in the next two sections that these problems are tractable if the *treewidth of the group intersection graph* is small.

## 3 A Dynamic Program for Best k-Group Approximation

The best $k$-group approximation problem is the discrete optimization problem

$$\arg\min_x \|y - x\|_2^2 \text{ such that } C_{\mathcal{G}}(\mathrm{supp}(x)) \leq k, \qquad (8)$$

where the goal is to compute the projection of a vector $y$ onto a union of subspaces each defined by a subcollection of $k$ groups. The solution to (8) has the form

$$x_i' = \begin{cases} y_i & i \text{ in chosen support}, \\ 0 & \text{otherwise}. \end{cases}$$

As mentioned above, Baldassarre et al. [3] show that this problem is NP-hard. They provide a dynamic program that acts on the *group intersection graph* and focus specifically on the case where this graph is a tree, obtaining a $O(m^2 k + n)$ dynamic programming algorithm. In this section, we also start by using group intersection graphs, but instead focus on the *tree decomposition* of this graph, which yields a more general approach.

### 3.1 Group Intersection Graphs and Tree Decompositions

We can represent the interactions between the different groups using an intersection graph, which is an undirected graph $I_{\mathcal{G}} = (\mathcal{G}, E_{\mathcal{G}})$ in which each vertex denotes a group and two groups are connected if and only if they overlap. For example, if the collection of groups is $\{\{1, 2, 3\}, \{3, 4, 5\}, \{5, 6, 7\}, \dots\}$, then the intersection graph is simply a chain. If each group corresponds to a parent and all its children in a rooted tree, the intersection graph is also a tree.

The group intersection graph highlights the dependencies between different groups. Algorithms for this problem need to be aware of how picking one group may affect the choice of another, connnected group. (If the groups do not overlap, then no groups are connected and a simple greedy approach suffices.) A *tree decomposition* of $I_{\mathcal{G}}$ is a more precise way of representing these dependencies.

We provide the definition of tree decompositions and treewidth below and illustrate the core ideas in Figure 2. Tree decompositions are a fundamental tool in parametrized complexity, leading to efficient algorithms if the parameter in question is small. See [7, 8] for a a more comprehensive overview

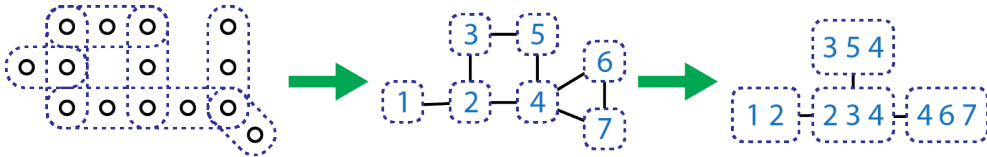

Figure 2: From groups to a group intersection graph to a tree decomposition of width 2.

A tree decomposition of $(V, E)$ is a tree $\mathcal{T}$ with vertices $\mathcal{X} = \{X_1, X_2, \ldots, X_N\}$, satisfying the following conditions: (1) Each $X_i$ is a subset of $V$, and the union of all sets $X_i$ gives $V$. (2) For every edge $(v, w)$ in $E$, there is a vertex $X_i$ that contains both $v$ and $w$. (3) For each $v \in V$, the vertices that contain $v$ form a connected subtree of $T$. The *width* of a tree decomposition is $\max_i |X_i| - 1$, and the *treewidth* of a graph (denoted by $tw$) is the smallest width among all tree decompositions. The tree decomposition is not unique, and there is always a tree width with number of nodes $|\mathcal{X}| \leq |V|$ (see for example Lemma 10.5.2 in [7]). Henceforth, we will assume that $|\mathcal{X}| \leq m$.

The treewidth $tw$ is modest for many types of graphs. For example, the treewidth is bounded for the tree ($tw = 1$), the cycle ($tw = 2$), and series-parallel graphs ($tw = 2$). Computing tree decompositions with optimal width for these graphs can be done in linear time. On the other hand, the grid graph has large treewidth ($tw \approx \sqrt{n}$) and checking if a graph has $tw \leq k$ is NP-complete[1].

## 3.2 A Dynamic Program for Tree Decompositions

Given a collection of groups $\mathcal{G}$, a corresponding tree decomposition $\mathcal{T}(\mathcal{G})$ of the group intersection graph, and a vector $y \in \mathbb{R}^n$, we provide a dynamic program for problem (8), the best $k$-group approximation of $y$.

The tree decomposition has several features that we can exploit. The tree structure provides a natural order for processing the vertices, which are subcollections of groups. Properties (1) and (2) yield a natural way to map elements $i \in [n]$ onto vertices in the tree, indicating when to include $y_i$ in the process. Finally, the connected subtree corresponding to each group $G$ as a result of property (3) means that we only need to keep explicit information about $G$ for that part of the computation.

The high-level view of our approach is described below. Details appear in the supplementary material.

**Preprocessing:** For each $i \in [n]$, let $\mathcal{G}_{(i)}$ denote the set of all groups that contain $i$. We have three data structures: A and V, which are both indexed by $(X, Y)$, with $X \in \mathcal{X}$ and $Y \subseteq X$; and T, which is indexed by $(X, Y, s)$, with $s \in \{0, 1 \ldots, k\}$.

1. Root the tree decomposition and process the nodes from root to leaves: At each node $X$, add an index $i \in [n]$ to $\mathtt{A}(X, \mathcal{G}_{(i)})$ if $i$ is unassigned and $\mathcal{G}_{(i)} \subseteq X$.

2. Set $\mathtt{V}(X, Y) \leftarrow \sum \{y_i^2 : i \in \mathtt{A}(X, \mathcal{G}_{(i)}), Y \cap \mathcal{G}_{(i)} \neq 0\}$.

**Main Process:** At each vertex $X$ in the tree decomposition, we are allowed to pick groups $Y$ to include in the support of the solution. The $s$ term in $\mathtt{T}(X, Y, s)$ indicates the current group sparsity "budget" that has been used. Proposition 3.2 below gives the semantic meaning behind each entry in $\mathcal{T}$. We process the nodes from the leaves to the root to fill $\mathcal{T}$. At each step, the entries for node $X_p$ will be updated with information from its children.

The update for a leaf $X_p$ is simply $\mathtt{T}(X_p, Y_p, s) \leftarrow \mathtt{V}(X_p, Y_p)$ if $|Y_p| = s$. If $|Y_p| \neq s$, we mark $\mathtt{T}(X_p, Y_p, s)$ as invalid. For non-leaf $X_p$, we need to ensure that the groups chosen by the parent and the child are compatible. We ensure this property via constraints of the form $Y_c \cap X_p = Y_p \cap X_c$. For a single child $X_c$ we have

$$\mathtt{T}(X_p, Y_p, s) \leftarrow \max_{Y: Y \cap X_p = Y_p \cap X_c} \left\{ \mathtt{T}(X_c, Y, s - s_0) : |Y_p \cap \overline{X_c}| = s_0 \right\} + \mathtt{V}(X_p, Y_p), \qquad (9)$$

and finally for $X_p$ multiple children $X_{c(1)}, \ldots, X_{c(d)}$ of $X_p$, we set $\mathtt{T}(X_p, Y_p, s)$ as

$$\max_{\substack{Y_i : Y_i \cap X_p = Y_p \cap X_{c(i)} \\ \text{for each } i}} \left\{ \sum_{\sum_{i=1}^d s_i = s - s_0} \mathtt{T}(X_{c(i)}, Y_i, s_i) : \left| Y_p \cap \overline{\bigcup_{i \in [d]} X_{c(i)}} \right| = s_0 \right\} + \mathtt{V}(X_p, Y_p). \quad (10)$$

After making each update, we keep track of which $Y_i$ was used for each of the children for $\mathcal{T}(X_p, Y_p, s)$. This allows us to backtrack to recover the solution after $\mathcal{T}$ has been filled.

The next lemma and proposition prove the correctness of this dynamic program. The lemma follows from the fact that every clique in a graph is contained in some node in any tree decomposition, while the proposition from induction from the leaf nodes.

**Lemma 3.1.** *Every index in $[n]$ is assigned in the first preprocessing step.*

**Proposition 3.2.** *For a node $X$, let $y_X$ be the $y$ vector restricted to just the indices $i$ assigned to nodes below and including $X$. Each entry $\mathtt{T}(X, Y, s)$ is the squared $\ell_2$-norm of the best projection of $y_X$, subject to the fact that besides the groups in $Y$, at most $s - |Y|$ are allowed to be used.*

We now prove the time complexity of this algorithm. Proposition 3.4 describes the time complexity of the update when there are many children. It uses the following simple lemma about *max-convolutions*. Computing the other updates is straightforward.

**Lemma 3.3.** *The max-convolution $f$ between two concave functions $g_1, g_2 : \{0, 1, \ldots, k\} \to \mathbb{R}$, defined by $f(i) := \max_j \{g_1(j) + g_2(i - j)\}$, can be computed in $O(k)$ time.*

**Proposition 3.4.** *The update* (10) *for a fixed $X_p, Y_p$ across all values $s \in \{0, 1, \ldots, k\}$ can be implemented in $O(2^{O(tw)} \cdot dk)$ time.*

Combining timing and correctness results gives us the desired algorithmic result. This approach significantly improves on the results of Baldassarre et al. [3]. Their approach is specific to groups whose intersection graph is a tree and uses $O(m^2 k + n)$ time.

**Theorem 3.5.** *Given $\mathcal{G}$ and a corresponding tree decomposition $\mathcal{T}_{\mathcal{G}}$ with treewidth $tw$, projection onto the corresponding $k$-group model can be done in $O(2^{O(tw)} \cdot (mk + n))$ time. When the group intersection graph is a tree, the projection takes $O(mk + n)$ time.*

## 4 Extended Formulations from Tree Decompositions

Here we model explicitly the unit ball of LG($k$) (5) and LGS-OWL (7). The principles behind this formulation are very similar to the dynamic program in the previous section.

We first consider the latent $k$-group support norm, whose atoms are

$$\left\{ x : \|x\|_p \leq 1, \ \mathrm{supp}(x) \subseteq \bigcup_{G \in \mathcal{G}_k} G \ \text{for some subset } \mathcal{G}_k \subseteq \mathcal{G} \text{ with } k \text{ groups} \right\}.$$

The following process describes a way of selecting an atom; our extended formulation encodes this process mathematically. We introduce variables $b$, which represent the $\ell_p$ budget at a given node, choice of groups, and group sparsity budget. We start at the root $X_r$, with $\ell_p$ budget of $\mu$ and group sparsity budget of $k$:

$$\sum b^{(X_r, Y, k - |Y|)} \leq \mu. \tag{11}$$

We then start moving towards the leaves, as follows.

1. Suppose we have picked some the groups at a node. Assign some of the $\ell_p$ budget to the $x_i$ terms, where the index $i$ is compatible with the node and the choice of groups.

2. Move on to the child and pick the groups we want to use, considering only groups that are compatible with the parent. Debit the group budget accordingly. If there are multiple children, spread the $\ell_p$ and group budgets among them before picking the groups.

The first step is represented by the following relations. Intermediate variables $z$ and $a$ are required to ensure that we spread the $\ell_p$ budget correctly among the valid $x_i$.

$$b^{(X,Y,s)} \geq \|(z^{(X,Y,s)}, u^{(X,Y,s)})\|_p, \tag{12}$$

$$z^{(X,Y,s)} \geq \|\{a^{(X,(Y,Y'),s)} : Y' \cap Y \neq \emptyset\}\|_p, \tag{13}$$

$$a^{(X,Y',s)} \leq \sum_{Y \subseteq X} a^{(X,(Y,Y'),s)}, \tag{14}$$

$$\|x_{\mathtt{A}(X,Y)}\|_2 \leq \sum_{\mathtt{A}(X,Y) = \mathtt{A}(X,Y')} \sum_{s=0}^{k} a^{(X,Y',s)}. \tag{15}$$

The second step is represented by the following inequality in the case of a single child.

$$u^{(X_p, Y_p, s)} \geq \sum \left\{ b^{(X_c, (Y_p, Y), s - s_0)} : Y \cap X_p = Y_p \cap X_c, |Y_p \cap \overline{X_c}| = s_0 \right\}. \tag{16}$$

When there are multiple children, we need to introduce more intermediate variables to spread the group budget correctly. The technique here is similar to the one used in the proof of Proposition 3.4; we defer details to the supplementary material. In both cases, we need to collect the budgets that have been sent from each $Y_p$:

$$b^{(X_c, Y, s)} \leq \sum_{Y_p} b^{(X_c, (Y_p, Y), s)}. \tag{17}$$

Those $b$ variables unreachable by the budget transfer process are set to 0. Our main theorem about the correctness of the construction in this section follows from the fact that when $\mu = 1$, every extreme point with nonzero $x$ in our extended formulation is an atom of the corresponding $\mathrm{LG}(k)$.

**Theorem 4.1.** *We can model the set $\Omega_{\mathrm{LG}(k)}(x) \leq \mu$ using $O(2^{O(tw)} \cdot (mk^2 + n))$ variables and inequalities in general. When the tree decomposition is a chain, $O(2^{O(tw)} \cdot (mk + n))$ suffices.*

For the unit ball of $\Omega_{\mathrm{LGS\text{-}OWL}}$, we can exploit the fact that the atoms of $\Omega_{\mathrm{LGS\text{-}OWL}}$ are obtained from $\Omega_{\mathrm{LG}(k)}$ across different $k$ at different scales. Instead of using the inequality (11) at the node, we have

$$\sum_{Y \subseteq X_r} h(k)^{1/q} b^{(X_r, Y, k - |Y|)} \leq \mu,$$

which leads to a program of size $O(2^{O(tw)} \cdot (m^2 + n))$ for chains and $O(2^{O(tw)} \cdot (m^3 + n))$ for trees.

## 5 Empirical Observations and Results

The extended formulations above can be implemented in modeling software such as CVX. This may incur a large processing overhead, and it is often faster to implement these directly in a convex optimization solver such as Gurobi or MOSEK. Use of the $\ell_\infty$-norm leads to a linear program which can be significantly faster than the second-order conic program that results from the $\ell_2$-norm.

We evaluated the performance of $\mathrm{LG}(k)$ and LGS-OWL on linear regression problems $\min_x \frac{1}{2} \|y - Ax\|^2 + \lambda \Omega(x)$. In the scenarios considered, we use the latent group lasso as a baseline. We test both the $\ell_2$ and $\ell_\infty$ variants of the various norms. Following [13] (which descrbes GrOWL), we consider two different types of weights for LGS-OWL. The linear variant sets $w_i = 1 - (i-1)/n$ for $i \in [n]$, whereas in the spike version, we set $w_1 = 1$ and $w_i = 0.25$ for $i = 2, 3, \ldots, n$. The regularization term $\lambda$ was chosen by grid search over $\{10^{-2}, 10^{-1.95}, \ldots, 10^4\}$ for each experiment.

The metrics we use are support recovery and estimation quality. For the support recovery experiments, we count the number of times the correct support was identified. We also compute the root mean square (RMSE) of $\|x - x^*\|_2$ (estimation error).[2]

We had also tested the standard lasso, elastic net, and $k$-support and OWL norms, but these norms performed poorly. In our experiments they were not able to recover the exact correct support in any run. The estimation performance for the $k$-support norms and elastic net were worse than the corresponding latent group lasso, and likewise for OWL vs. LGS-OWL.

**Experiments.** We used 20 groups of variables where each successive group overlaps by two elements with the next [10, 14]. The groups are given by $\{1, \ldots, 10\}, \{9, \ldots, 18\}, \ldots, \{153, \ldots, 162\}$. For the first set of experiments, the support of the true input $x^*$ are a cluster of five groups in the middle of $x$, with $x_i = 1$ on the support. For the second set of experiments, the original $x$ is supported by the two disjoint clusters of five overlapping groups each, with $x_i = 2$ on one cluster and $x_i = 3$ on the other.

Each entry of the $A$ matrix is chosen initially to be i.i.d. $\mathcal{N}(0, 1)$. We then introduce correlations between groups in the same cluster in $A$. Within each cluster of groups, we replicate the same set of columns for each group in the non-overlapping portions of the group (that is, every pair of groups in a cluster shares at least 6 columns, and adjacent groups share 8 columns). We then introduce noise by adding i.i.d. elements from $\mathcal{N}(0, 0.05)$ so that the replications are not exact. Finally, we generate $y$ by adding i.i.d. noise from $\mathcal{N}(0, 0.3)$ to each component of $Ax^*$.

We present support recovery results in Figure 3 for the $\ell_2$ variants of the norms which perform better than the $\ell_\infty$ versions, though the relative results between the different norms hold. In the appendix we provide the graphs for support recovery and estimation quality as well as other observations.

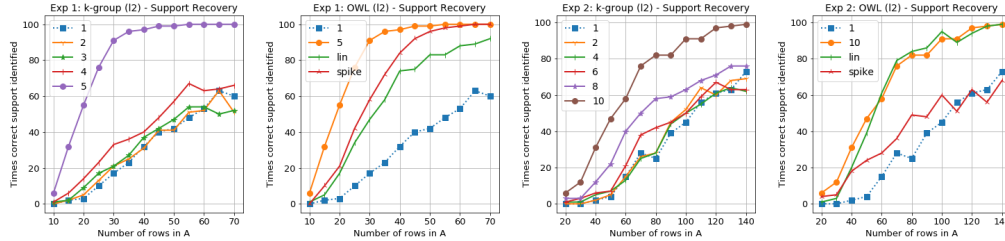

Figure 3: Support recovery performance as number of measurements (height of $A$) increases. The vertical axis indicates the number of trials (out of 100) for which the correct support was identified. The two left graphs correspond to the first configuration of group supports (five groups), while the others to the second configuration (ten groups). Each line represents a different method. In the first and third graphs, we plot LG($k$) for different values of $k$, increasing from 1 to the "ground truth" value. Note that $k = 1$ is exactly the latent group lasso. In the second and fourth graphs, we plot LGS-OWL for the different choices of weights $w_i$ discussed in the text.

Our methods can significantly outperform latent group lasso in both support recovery and estimation quality. We provide a summary below and more details are provided in the supplementary.

We first focus on support recovery. There is a significant jump in performance when $k$ is the size of the true support. Note that exceeding the ground-truth value makes recovery of the true support impossible in the presence of noise. For smaller values of $k$, the results range from slight improvement (especially when $k = 4$ or $k = 8$ in the first and second experiments respectively) to mixed results (for large number of rows in $A$ and small $k$). The LGS-OWL norms can provide performance almost as good as the best settings of $k$ for LG($k$), and can be used when the number of groups is unknown. We expect to see better performance for well-tuned OWL weights. We see similar results for estimation performance. Smaller values of $k$ provide little to no advantage, while larger values of $k$ and the LGS-OWL norms can offer significant improvement.

## 6 Discussion and Extensions

We introduce a variant of the OWL norm for overlapping groups and provide the first tractable approaches for this and the latent $k$-group support norm (via extended formulations) under a bounded treewidth condition. The projection algorithm for the best $k$-group sparse approximation problem generalizes and improves on the algorithm by Baldassarre et al. [3]. Numerical results demonstrate that the norms can provide significant improvement in support recovery and estimation.

A family of graphs with many applications and large treewidth is the set of grid graphs. Groups over collections of adjacent pixels/voxels lead naturally to such group intersection graphs, and it remains an open question whether polynomial time algorithms exist for this set of graphs. Another venue for research is to derive and evaluate efficient approximations to these new norms.

It is tempting to apply recovery results on the latent group lasso here, since LG($k$) can be cast as a latent group lasso instance with groups $\{G' : G' \text{ is a union of up to } k \text{ groups of } \mathcal{G}\}$. The consistency results of [10] only applies under the strict condition that the target vector is supported exactly by a unique set of $k$ groups. The Gaussian width results of [16] do not give meaningful bounds even when the groups are disjoint and $k = 2$. Developing theoretical guarantees on the performance of these methods requires a much better understanding of the geometry of unions of overlapping groups.

We can easily extend the dynamic program to handle the case in which we want both $k$-group sparsity, and overall sparsity of $s$. For tree-structured group intersection graphs, our dynamic program has time complexity $O(mks + n \log s)$ instead of the $\tilde{O}(m^2ks^2 + mn)$ by [3]. This yields a variant of the above norms that again has a similar extended formulations. These variants could be employed as an alternative to the sparse overlapping set LASSO by Rao et al. [14]. We leave this to future work.

**Acknowledgements** This work was supported by NSF award CMMI-1634597, ONR Award N00014-13-1-0129, and AFOSR Award FA9550-13-1-0138.

## Footnotes

[1]Nonetheless, there is significant research on developing exact and heuristic tree decomposition algorithms. There are regular competitions for better implementations [6, `pacechallenge.wordpress.com`].

[2]It is standard in the literature to compute the RMSE of the prediction or estimation quality. RMSE metrics are not ideal in practice since we should "debias" $x$ to offset shrinkage due to the regularization term.

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
