[Supplementary Material · ef_full.pdf]

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

# A Hardness Results

To formally define NP-hardness, we need to be clear about what the inputs are. These are $y \in \mathbb{R}^n$ and the collection of groups $\mathcal{G}$ for both norms, and in the case of the $\Omega_{\mathrm{LG}(k)}$ we include the $k$ parameter. For the pOWL norm, we assume that the function $h$ in the definition of the norm is strictly concave.

**Theorem 2.1.** *The following problems are NP-hard for both $\Omega_{\mathrm{LG}(k)}$ and $\Omega_{\mathrm{LGS\text{-}OWL}}$ when $p > 1$:*

$$Compute\ \Omega(y), \qquad \text{(norm computation)}$$

$$\arg\min_{x \in \mathbb{R}^n} 1/2\|x - y\|_2^2 \ such\ that\ \Omega(x) \leq \mu, \qquad \text{(projection operator)}$$

$$\arg\min_{x \in \mathbb{R}^n} 1/2\|x - y\|_2^2 + \lambda\Omega(x). \qquad \text{(proximal operator)}$$

*Proof.* We focus on the LG($k$) case first, and will describe how to extend this argument to the LGS-OWL norm after. We use a reduction from the NP-complete *vertex cover problem* to show this. Given an undirected graph $(V, E)$, a vertex cover $W \subset V$ is a collection of vertices such that every edge touches at least one point in $W$. The goal of the vertex cover problem is to determine if there exists a vertex cover of size at most $k$.

We can map each graph $(V, E)$ into a collection of groups $\mathcal{G}$ over the ground set $E$. Each vertex $v$ represents a group that contains all edges $e$ that touch the node.

We will show that the unit ball of $\Omega_{\mathrm{LG}(k)}$ is unit $\ell_p$ ball if and only if there is a vertex cover of size $k$, and if there is not then it is a strict subset of the $\ell_p$ ball. Note that we can include every single index in some choice of $k$ groups if and only if we can find a vertex cover of size $k$. This means that the $\Omega_{\mathrm{LG}(k)}$ is the $\ell_p$-norm if there is a vertex cover of size $k$. On the other hand, suppose there is no $k$ vertex cover. Then, there is no atom of unit norm 1 that has only nonzero indices, and any strict convex combination of atoms necessarily leads to points with unit norm less than 1 for $p > 1$.

We will reduce the vertex cover problem to each of the three problems above for LG($k$). Consider the point $y = c\mathbf{1}$, where $c$ scales the all-ones vector $\mathbf{1}$ such that $\|y\|_p = 1$. There is a vertex cover of size $k$ if and only if the norm $\Omega_{\mathrm{LG}(k)}(y)$ is equal to one, or if and only if the vector $y$ projects back onto itself.

The proximal operator requires more care. Firstly, the minimizer in the case where the $\Omega_{\mathrm{LG}(k)}$ is the $\ell_p$ norm is given by $x^*$ where

$$x_i^* + \lambda(x_i^*/\|x^*\|_p)^{p-1} = c$$

so all the entries in $x^*$ are the same. If the returned $x$ does not have homogeneous entries, then we know that there is no vertex cover of size $k$. So we only need to focus on the homogeneous $x$ case, a one dimensional problem. The objective value is now a quadratic function given by

$$(1/2)n(z - c)^2 + \lambda\Omega((z, z, \ldots, z))$$

for $z \in \mathbb{R}$ and note that is $\Omega_{\mathrm{LG}(k)}$ is the $\ell_p$ norm the objective is $1/2n(z - c)^2 + \lambda\sqrt[p]{n}z$, a simple quadratic form. If $\Omega_{\mathrm{LG}(k)}$ is not the $\ell_p$ norm, then $\Omega((z, z, \ldots, z)) = c'z < \sqrt[p]{n}z$ for some constant $c'$ and hence the solution is different.

The size of the smallest vertex cover is $C_{\mathcal{G}}([n])$. Using the fact that $h$ is monotonically increasing, we can express the unit ball of LGS-OWL as

$$\mathrm{conv}\left(\bigcup_{i=1}^{C_{\mathcal{G}}([n])} \left\{x \in \mathbb{R}^n : \|x\|_p \leq h(i)^{-1/q}, C_{\mathcal{G}}(\mathrm{supp}(x)) = i\right\}\right),$$

which ignores all the LG($k$) balls corresponding to $k > C_{\mathcal{G}}([n])$. For the same point $y$ as above, we have $\Omega_{\mathrm{LGS\text{-}OWL}}(y) = h(k)^{1/q}$ (follows from concavity of $h$). The proofs for the LGS-OWL norm then follows from similar arguments. □

# B Dynamic Programming Examples

In the first example, we have four indices and three groups. All three groups share exactly one index – $G_1 = \{1, 2\}$, $G_2 = \{2, 3\}$, and $G_1 = \{2, 4\}$. The only valid tree decomposition has all three groups in a single node, so $\mathcal{X} = \{\{1, 2, 3\}\}$. Let $X$ denote the set of groups $\{1, 2, 3\}$.

Figure 4: Three groups on four indices forming a 'star'.

We assign the indices to collections of groups as follows.
$$A(X, \{1\}) = \{1\}, A(X, X) = \{2\}, A(X, \{2\}) = \{3\}, A(X, \{3\}) = \{4\}.$$
We can now initialize the V table as follows
$$V(X, \{1\}) = v_1^2 + v_2^2, V(X, \{2\}) = v_2^2 + v_3^2, V(X, \{3\}) = v_3^2 + v_4^2$$
$$V(X, \{1, 2\}) = v_1^2 + v_2^2 + v_3^2, V(X, \{1, 3\}) = v_1^2 + v_2^2 + v_4^2, V(X, \{2, 3\}) = v_2^2 + v_3^2 + v_4^2$$
$$V(X, X) = v_1^2 + v_2^2 + v_3^2 + v_4^2.$$
To determine the T table, it suffices to note that $T(X, Y, |Y|) = V(X, Y)$.

We now consider the slightly less trivial case of a three group chain represented in Figure 5: The

Figure 5: Three groups on five indices forming a chain.

groups here are $G_1 = \{1, 2\}$, $G_2 = \{2, 3, 4\}$, and $G_3 = \{4, 5\}$. There are two valid tree decompositions, but we consider the one where we have two nodes $X_1 = \{1, 2\}$ and $X_2 = \{2, 3\}$. Let $X_1$ be the root node in the decomposition. The assignment table is
$$A(X_1, \{1\}) = \{1\}, A(X_1, \{1, 2\}) = \{2\}, A(X_2, \{2\}) = \{3\}$$
$$A(X_2, \{2, 3\}) = \{4\}, A(X_2, \{3\}) = \{5\},$$
and the value table V is
$$V(X_1, \{1\}) = v_1^2 + v_2^2, V(X_1, \{2\}) = v_2^2 + v_3^2, V(X_1, \{1, 2\}) = v_1^2 + v_2^2 + v_3^2,$$
$$V(X_2, \{2\}) = v_4^2, V(X_2, \{3\}) = v_4^2 + v_5^2, V(X_2, \{2, 3\}) = v_4^2 + v_5^2.$$
We now start forming the main table T. The leaf node is $X_2$, and we have
$$T(X_2, \{2\}, 1) = v_4^2, T(X_2, \{3\}, 1) = v_4^2 + v_5^2, T(X_2, \{2, 3\}, 2) = v_4^2 + v_5^2.$$
Computing the main table values at the root node requires us to use the update rule (9), and we have
$$T(X_1, \{1\}, 1) \leftarrow V(X_1, \{1\}) = v_1^2 + v_2^2,$$
$$T(X_1, \{1\}, 2) \leftarrow T(X_2, \{3\}, 1) + V(X_1, \{1\}) = v_1^2 + v_2^2 + v_4^2 + v_5^2,$$
$$T(X_1, \{2\}, 1) \leftarrow T(X_2, \{2\}, 1) + V(X_1, \{2\}) = v_2^2 + v_3^2 + v_4^2,$$
$$T(X_1, \{1, 2\}, 2) \leftarrow T(X_2, \{2\}, 1) + V(X_1, \{1, 2\}) = v_1^2 + v_2^2 + v_3^2 + v_4^2,$$
$$T(X_1, \{1, 2\}, 3) \leftarrow T(X_2, \{2, 3\}, 2) + V(X_1, \{1, 2\}) == v_1^2 + v_2^2 + v_3^2 + v_4^2 + v_5^2.$$

## C  Dynamic Programming Proofs

**Lemma 3.1.** *Every index in $[n]$ gets assigned in the first preprocessing step.*

*Proof.* Let $\mathcal{G}_{(i)}$ denote the groups that index $i$ is contained in. To prove this claim, it suffices to show that there is a vertex $X$ in the tree decomposition that contains $\mathcal{G}_{(i)}$ for each $i \in [n]$.

A classic result in tree decompositions is that every clique in the original graph is contained in some vertex of the tree decomposition (see for example Lemma 10.1.1 in Downey and Fellows [8]). Since the groups in $\mathcal{G}_{(i)}$ form a clique in the intersection graph, our result immediately follows. $\qquad\square$

**Proposition 3.2.** *For a node $X$, let $y_X$ be the $y$ vector restricted to just the indices $i$ assigned to nodes below and including $X$. Each entry $\mathtt{T}(X, Y, s)$ is the squared $\ell_2$-norm of the best projection of $y_X$, subject to the fact that besides the groups in $Y$, at most $s - |Y|$ are allowed to be used.*

*Proof.* We prove this by induction from the leaves up. The claim holds for the leaves since we use the right group sparsity budget depending on the number of groups we choose to be active, and the $\mathtt{V}$ table ensures that we add all the squared $y_i$ values that we can add according to the groups chosen.

Consider the single child update (9). Suppose we have decided to pick the set of groups $Y_p$ at node $X_p$, and we are allowed to have spent a group sparsity budget of $s$. We are allowed to add all the $y_i$ values corresponding to our choice of $Y_p$. We now consider the choices we can make for the vertices below $Y_p$. We need to ensure that the choices of groups between $X_p$ and its immediate child $X_c$ are compatible, hence the $Y_c \cap X_p = Y_p \cap X_c$ restriction. We also need to make sure that we have the right sparsity level. Once we have picked subset for the child $Y_c$, then the choices of subsequent groups below is obvious: we pick the choice of groups that maximize the $\ell_2$-norm of the indices we have seen so far that are compatible with $Y_c$. The corresponding sum of squared $y_i$ values is exactly $\mathtt{T}(X_c, Y_c, s - s_0)$ by the induction hypothesis. Hence, the choice now is to maximize over all the possible compatible collections of groups $Y_c$ for the child.

For the update where we have multiple children (10), the same logic applies, except now over multiple vertices. We need to ensure that the group sparsity budget is correctly tallied up among all the children. We also need to make sure that the choices of all the groups are compatible not just between the single parent and their children, but also between the different children. For the latter problem, we exploit the third property of tree decompositions – any group $G$ that is contained in two different children must also be contained in the parent. This ensures that it is sufficient to check compatibility between the parent and each child without having to worrying about interactions across children (which could exponentially increase complexity). $\square$

**Lemma 3.3.** *The max-convolution $f$ between two concave functions $g_1, g_2 : \{0, 1, \ldots, k\} \to \mathbb{R}$ defined as $f(i) := \max_j \{g_1(j) + g_2(i - j)\}$ can be computed in $O(k)$ time.*

*Proof.* We first compute functions $g_1', g_2'$ where $g_k'(0) = g_k(0)$ and $g_k'(i) = g_k(i) - g_k(i-1)$. Since the $g$ functions are concave, this means that $g_k'$ functions are decreasing. If $g_1'(a)$ is larger than $g_2'(b)$, then $f(a + b - 1) \geq g_1(a) + g_2(b - 1) > g_1(a - 1) + g_2(b)$. This swapping argument implies that an algorithm that greedily picks $g_1'$ or $g_2'$ depending on which is larger gives us the optimal solution.

More formally, we start with $f(0) = g_1(0) + g_2(0)$ and repeat the following process. Let $i$ indicate the current index of $f$ and indices $j, k$ indicate the solution $f(i) = g_1(j) + g_2(k)$. We then consider $g_1'(j + 1)$ and $g_2'(k + 1)$. If the first term is larger, then $f(i + 1) = g_1(j) + g_2(k)$, otherwise $f(i + 1) = g_1(j) + g_2(k + 1)$. $\square$

**Proposition 3.4.** *The update* (10) *for a fixed $X_p, Y_p$ across all values $s \in \{0, 1, \ldots, k\}$ can be implemented in $O(2^{O(tw)} \cdot dk)$ time.*

*Proof.* It actually suffices to show that this can be done for $d = 2$ in time $O(2^{O(tw)} \cdot k)$ for the following reason: Suppose a parent $X_p$ has multiple children $X_{c(1)}, X_{c(2)}, \ldots, X_{c(d)}$. Then we can simply add a caretaker node $X_{p'}$ between $X_p$ and all their children $X_{c(2)}, \ldots, X_{c(d)}$ where $X_{p'}$ has all the same groups as $X_p$. This is still a valid tree decomposition and we can repeat the process.

For $d = 2$, $\mathtt{T}(X_p, Y_p, s)$ can be written as

$$\max_{\substack{Y_i : Y_i \cap X_p = Y_p \cap X_{c(i)} \forall i \\ |Y_p \cap \overline{\bigcup_{i \in [2]} X_{c(i)}}| = s_0 \\ s_1 + s_2 = s - s_0}} \left\{ \mathtt{T}(X_{c(1)}, Y_1, s_1) + \mathtt{T}(X_{c(2)}, Y_2, s_2) \right\} + \mathtt{V}(X_p, Y_p). \tag{18}$$

For each fixed $Y_p, Y_1$, and $Y_2$, we just need to make sure that $s_1 + s_2 = s - s_0$. To compute the maximizer over all $s$, this is equivalent to computing the max-convolution of $\mathtt{T}(X_{c(1)}, Y_1, \cdot)$ and $\mathtt{T}(X_{c(2)}, Y_2, \cdot)$. The concavity of these functions follows from the fact that the projection problem can be cast as a submodular problem with a cardinality constraint [3]. $\square$

**Theorem 3.5.** *Given $\mathcal{G}$ and a corresponding tree decomposition $\mathcal{T}_{\mathcal{G}}$ with treewidth $tw$, projection onto the corresponding $k$-group model can be done in $O(2^{O(tw)} \cdot (mk + n))$ time. When the group intersection graph is a tree, the projection takes $O(mk + n)$ time.*

*Proof.* The correctness of the approach follows from Proposition 3.2. The running time analysis from Proposition 3.4 give us a running time of 3.2 $O(2^{O(tw)} \cdot mk)$, since there are up to $m - 1$ children in the graph. This covers the complexity of filling the main table $\mathcal{T}$.

We now analyze the running time for filling the tables V and A. For A, at each vertex we consider groups we have not seen ($O(tw)$), and look at each element in these groups that have not been assigned ($O(tw \cdot n)$ work across all vertices, since each vertex is contained in up to $tw + 1$ groups). If the element can be assigned, we assign it. Once we assign an index $i$, we add $y_i^2$ to all relevant groups ($O(2^t w)$) in the V table. $\qquad\square$

# D  Extended Formulation

**Construction for Vertices with Multiple Children.** As with the proof of Proposition 3.4, we can assume that each node has at most two children. Then, the extended formulation analogue of (18) is formed by the following set of inequalities. The first two decides how we are splitting the sparsity budget of $s$ among both children:

$$u^{(X_p, Y_p, s)} \geq \sum_{\substack{Y_i: Y_i \cap X_p = Y_p \cap X_{c(i)} \forall i \\ |Y_p \cap \overline{\bigcup_{i \in [2]} X_{c(i)}}| = s_0 \\ s_1 \leq s - s_0}} u^{(X_{c(1)}, (Y_p, Y_{c(1)}, Y_{c(2)}), s - s_0, s_1)},$$

$$u^{(X_{c(1)}, (Y_p, Y_{c(1)}, Y_{c(2)}), s, s_1)} \geq \left\| \left( b^{(X_{c(1)}, (Y_p, Y_{c(1)}, Y_{c(2)}), s, s_1)}, b^{(X_{c(2)}, (Y_p, Y_{c(2)}, Y_{c(1)}), s, s - s_1)} \right) \right\|_p.$$

We now need to accumulate all the different $b$ variables

$$b^{(X_{c(i)}, (Y_p, Y_{c(i)}), s)} \leq \sum_{Y_{c(3-i)} \subseteq X_{c(3-i)}} b^{(X_{c(1)}, (Y_p, Y_{c(i)}, Y_{c(3-i)}), s)}$$

before applying the accumulation inequality 17 for the single child case.

Before we prove the main theorem in this section, we need the following lemma.

**Lemma D.1.** *When $\mu = 1$, every extreme point with nonzero $x$ in our extended formulation is an atom of the corresponding* $\mathrm{LG}(k)$.

*Proof.* For each summation term in the extended formulation, at most one variable (or pair in the case of (18)) will be active at any extreme point. In particular, this means that we pick exactly one set $Y$ for each vertex $X$ in inequalities (16) and (18) for assigning $\ell_p$ budget to the children. Satisfying each inequality with an $\ell_p$ term at equality ensures that the total $\ell_p$ budget that trickles down to each $x$ terms is precisely $\mu$. $\qquad\square$

The main result Theorem 4.1 then follows from convexity of the formulation, and the fact that the number of variables and inequalities are similar and on the order of $|X| \times 2^{O(tw)} \times k^2$.

# E  Additional Empirical Results and Other Observations

**Performance of $\ell_2$ vs. $\ell_\infty$ norm.** In terms of support recovery, the $\ell_2$ norm does consistently better than the $\ell_\infty$ norm and the difference is observed across all values of $k$ and types of LGS-OWL norms. However, when it comes to estimation performance the results depend a lot more on the choice of $k$ and the experiment. For the first experiment, $\ell_\infty$ ranges from slightly worse to much better than $\ell_2$, and the larger $k$ is, the bigger the improvement $\ell_\infty$ offers. This is possibly due to fact that the solution has constant values on the support, though this would not explain the support recovery results. On the other hand, in the second experiment where the solution is different across different correlated groups, we see that the $\ell_2$ norm is also better for estimation.

Figure 6: Experiment 1, Support Recovery. LG($k$) and LGS-OWL with $\ell_2$-norm and $\ell_\infty$-norm.

Figure 7: Experiment 2, Support Recovery. LG($k$) and LGS-OWL with $\ell_2$-norm and $\ell_\infty$-norm.

Figure 8: Experiment 1, Estimation RMSE. LG($k$) and LGS-OWL with $\ell_2$-norm and $\ell_\infty$-norm.

Figure 9: Experiment 2, Estimation RMSE. LG($k$) and LGS-OWL with $\ell_2$-norm and $\ell_\infty$-norm.

**OWL Observations.**   Whether the linear or spike version of OWL performs better depends on a large number of factors, including whether the $\ell_2$ or $\ell_\infty$ norm is used, the number of rows in $A$, and the particular experiment being used. There is no clear winner between the two.

The LGS-OWL results that we present in this work only consider two choices of weights. It is likely that a properly tuned choice of weights could further improve the performance. For example, [13] performed cross validation to pick the best parameters for the linear and spike variants of OWL. Another choice of weights would be based on the inverse CDF of the Gaussian [4], which has been shown to effectively control false discovery rate.

The extended formulation for LGS-OWL norms do not scale as well to the number of groups as the $\mathrm{LG}(k)$ norms, since we need to introduce variables that correspond to all sparsity budgets from 1 to $m$. As an alternative, we can 'truncate' the LGS-OWL norm by removing all variables corresponding to sparsity budgets above a threshold $T$. This is equivalent to removing the atoms $\mathrm{LG}(k)$ for $k \geq T$ in the atomic norm formulation (7). These new norms retain the empirical performance of LGS-OWL in our experiments if $T$ is sufficiently large.