[Reviews · NeurIPS 2017]

Reviewer 1



abstract This paper presents a unifying view of k-support/OWL norms and overlapping group-lasso norms. It results in two new families of norms, called "latent k-group-support" (LGS) norm and "latent group-smooth OWL" (LGS-OWL) norm). Computation of these norms is in general NP-hard. However, when the underlying groups has a special form characterised by a tree decomposition of the grouping, it is possible to compute these norms with known complexity. This complexity depends on the tree width, which is a measure of overlapping of the groups. A dynamic program relying on the tree decomposition paves the way for projecting onto the LGS and LGS-OWL norms. Some numerical examples of support recovery comparing the latent group-Lasso to these extensions with varying k are studied. comments The paper is well written, even though I found it rather technical since it is not exactely my area of expertise. I am impressed by the computational results and the dynamic program derived y the authors to compute the projection onto these new norms. Thus, on the computational and practical side, I think that there is a truly new contribution for the special topic of norms crafting. On the other hand, I am not completly convinced by the utility of these new norms compared to the existing OWL/k-support. In particular, in the numerical studies, it is obvious that knowning the number of latent group is very important for achieving good performance in terms of support recovery. This is not the case in practice, how do the authors deal with this issue? Beside, I think that the unifying papers of Obozinski and Bach already characterized a very large family of norms.

Reviewer 2



The paper adapts recent sparsity-inducing regularizers (pOWL, k-support norm) to the overlapping group setting. Two norms are derived from formulations of pOWL and k-support in the combinatorial penalty framework of Obozinski and Bach, where the "cardinal of the support" submodular function is replaced by the minimum number of groups required to cover the support, which is not submodular. The papers shows most related problems (norm computation, projection and proximal operator) to be NP-hard in general. In the case where a graph constructed from the group overlapping patterns has small treewidth, the papers shows that level sets of the norm can be modelled using a small number of linear and l_p cone constraints. Experiments are based this formulation using a general purpose convex solver. On synthetic data, the proposed norms are compared to latent group Lasso. The paper is well written. Sections 3.2 and 4 are a bit dense and dry, and could benefit from working out concrete examples, possibly in the supplemental material. The paper is interesting, and the proposed methods appear promising for overlapping group problems where one wants to stay within the realm of convex methods. Experiments on synthetic data are rigorous and convincing, although overall the experimental section is a bit short. More experiments, including on non-synthetic data, would be interesting. Remarks L180: "indices i assigned to nodes below and including X" is a bit confusing to me. Nodes are sets of groups, so I am assuming this means "indices i in the support of groups assigned to nodes below/in X"? L196: "the the unit ball" L242: is "..." missing in the 3rd sample group? L337: vertex -> vertices

Reviewer 3



Summary: This paper designs new norms for group sparse estimation. The authors extend the k-support norm and the ordered weighted norm to the group case (with overlaps). The resulting (latent) norms are unfortunately NP-hard to compute, though. The main contribution is an algorithm based on tree decomposition and dynamic programming for computing the best approximation (under the Euclidean norm) under group cardinality constraints. This algorithm improves the previous work by a factor of m (# of groups). A second contribution is an extended formulation for the proposed latent group norms. Such formulation can be plugged into standard convex programming toolboxes, as long as the underlying tree width is small. Some synthetic experiments are provided to justify the proposed norms (and their extended formulations). Overall this paper appears hard to follow, largely due to its technical nature. The organization is also a bit off: the utility of the dynamic programming algorithm in section 3 is largely unclear, and the experiment section did not verify this part at all. While being mostly a computational paper, strangely, the experiment section only compared the statistical performances of different norms, without conducting any computational comparison. While the technical results seem to be solid and improve previous work, it will be more convincing if more can be said on what kind of problems can benefit from the authors' results. With these much more sophisticated norms, one becomes easily overwhelmed by the fine mathematical details but somehow also gets more suspicious about their practical relevance (which unfortunately is not well addressed by the current experiments). Other comments: Lines 93-95: the parameter nu disappears in the convex envelope? Line 97: the weights need to be nonnegative. Line 343: this proof of Theorem 2.1 works only for 1 < p < infty? For instance, if p = 1, then Omega = |.|_1 always. Please fully justify the claim "Omega is the ell_p norm if and only if ...", as the experiments did use p=infty. Line 346: please justify "if lambda < 1, the solution is (1-lambda)y if and only if ..." Line 141: I do not see why the claim N <= |V| is correct. do you mean max_j |X_j| <= |V|? Experiments: should also include a set of experiments that favors the latent group norm, to see that if the proposed norms perform (much?) worse under an unfavorable condition. Also, please compare with the elastic net regularizer (in the correlated design case).